# Functional comparison of metabolic networks across species

Charlotte Ramon [1,2] & Jörg Stelling [1] ✉

Metabolic phenotypes are pivotal for many areas, but disentangling how evolutionary history and environmental adaptation shape these phenotypes is an open problem. Especially for microbes, which are metabolically diverse and often interact in complex communities, few phenotypes can be determined directly. Instead, potential phenotypes are commonly inferred from genomic information, and rarely were model-predicted phenotypes employed beyond the species level. Here, we propose sensitivity correlations to quantify similarity of predicted metabolic network responses to perturbations, and thereby link genotype and environment to phenotype. We show that these correlations provide a consistent functional complement to genomic information by capturing how network context shapes gene function. This enables, for example, phylogenetic inference across all domains of life at the organism level. For 245 bacterial species, we identify conserved and variable metabolic functions, elucidate the quantitative impact of evolutionary history and ecological niche on these functions, and generate hypotheses on associated metabolic phenotypes. We expect our framework for the joint interpretation of metabolic phenotypes, evolution, and environment to help guide future empirical studies.

Metabolic reactions as well as entire metabolic networks establish function by yielding phenotypes in terms of metabolic flux distributions inside the cell and in the cell's interaction with the environment. Such metabolic phenotypes of potentially complex cell communities impact many areas, including biogeochemical cycles[1] and human health[2]. Understanding the drivers of metabolic functional diversity requires disentangling links between metabolic gene repertoires, realized metabolic phenotypes, taxonomy to represent evolutionary history, and environmental characteristics. However, inferring these links, and ultimately determining how all factors combined shape metabolic phenotypes, is an open problem[3]. One challenge is that, especially in complex microbial communities, few phenotypes can be determined directly[4]. Instead, potential phenotypes are often inferred from genomic information[5]. Analyses of the global ocean microbiome illustrate common approaches based on metagenomics data: to infer metabolic functions from gene repertoires[6], or to use species-level functional annotations[7], which are then associated with taxonomy and environment. However, this does not consider interdependencies of

genes, cellular networks their products establish, and phenotypes. Since network context shapes gene functions, and the whole network generates metabolic phenotypes. genetic epistasis[8] and variable trait relations along the phylogeny[3] indicate a need to incorporate interactions in cellular networks.

Genome-scale metabolic network models (GSMs) make these dependencies explicit. They can reliably predict metabolic phenotypes[9], their topological analysis can predict environments[10,11], and they were instrumental in analyzing enzyme evolution[12], but all for single species. However, there are only a few studies that employed model-predicted phenotypes beyond the species level[13–15]. A recent, comparative GSM-based study of bacterial phenotype evolution[13] did not make links to genotype or environment, while other comparative studies linked specific (minimal)[15] or generic (sampled)[14] environments to species and ecological relations, but not to detailed metabolic functions. To bridge the corresponding gaps, here we exploit the concept that genotype–phenotype relationships connect differences at the genomic level (and in environments) with differences in

[1]Department of Biosystems Science and Engineering and SIB Swiss Institute of Bioinformatics, ETH Zurich, 4058 Basel, Switzerland. [2]Ph.D. Program Systems Biology, Life Science Zurich Graduate School, Zurich, Switzerland. ✉e-mail: joerg.stelling@bsse.ethz.ch

phenotypes[16]. Specifically, we quantify how perturbations in enzyme-catalyzed reactions affect metabolic fluxes to compare identical biochemical reactions and subsystems across species with varying metabolic network structures.

## Results

### Functional comparisons via sensitivities

Our framework uses structural sensitivity analysis[17] to characterize perturbation effects in metabolic networks (Fig. 1a). It uses only the network structure (that is, the stoichiometry of metabolic reactions) to assess how perturbations of metabolic fluxes propagate through the network. Specifically, structural sensitivities measure the predicted adjustments to all fluxes required to return a network to steady-state when one or more reactions in the network is perturbed. The predictions assume that cells tend to minimally redistribute fluxes upon a perturbation; this assumption allowed more accurate predictions of bacterial growth rates upon a genetic knockout[18]. To link genes and enzymes to fluxes, we use gene-protein-reaction mappings, which are a common component of GSMs[19] (see Methods for details). Here, we compute absolute sensitivities, which are obtained analytically (see Methods) and do not assume a specific operating state of the network or a specific environment[17]. These sensitivities are considered to capture adjustments to infinitesimal perturbations; they are valid unless a network's operating state is exactly equal to specific constraints such as the availability of nutrients in a specific environment, which is unlikely. To compare two common reactions (reactions with identical biochemical formula) in two GSMs, we correlate the sensitivities of all common reactions to perturbations of these two reactions (Fig. 1a and Methods). Correspondingly, we use 'function' ('functional similarity') in the sense of (similarity of) flux responses to perturbations of the common reactions.

We first evaluated if Pearson correlations of sensitivities provide information on network similarity that is different from measures based on reaction presence / absence (the metabolic repertoire) such as the Jaccard index. For this purpose, we quantified the similarities of the neighborhoods of each common reaction in the *Escherichia coli* and *Bacillus subtilis* networks. Sensitivity correlations and Jaccard indices for 1-neighborhoods do not correlate (Fig. 1b, $R^2 = 0.003$). In particular, many reactions have a low Jaccard index, but a high sensitivity correlation because sensitivities account for the whole graph's response to a perturbation; they can distribute over large graph distances (Fig. S1a). Jaccard indices do not capture this even when considering the 2-neighborhood of reactions (Fig. S1b).

To illustrate how sensitivity correlations capture the effects of network context on enzyme function, we consider ornithine carbamoyl transferase, which operates in the structurally similar, but not identical arginine biosynthesis pathways of *E. coli* and *B. subtilis* (Fig. 1c). Adding the two missing reactions to the *E. coli* GSM increases context similarity, increasing the sensitivity correlation from 0.61 to 0.74 (Fig. 1d). Sensitivity correlations can also pinpoint known structural differences between two GSMs. For example, the sensitivities to perturbing the reaction 5-amino-6-(5-phosphoribosylamino)uracil reductase in the riboflavin pathway are uncorrelated between *B. subtilis* and *E. coli* ($R^2 = 0$) because *B. subtilis* can adapt by active riboflavin transport across its membrane, but *E. coli* lacks this transport[20,21]. Correspondingly, the correlation increased to 0.71 when augmenting the *E. coli* GSM with riboflavin exchange and transport (Fig. S1c). These examples and the wide spread of sensitivity correlations (Fig. 1d) suggest that our measure is sufficiently fine-grained to differentiate metabolic functions, in contrast to comparisons of metabolic repertoires alone.

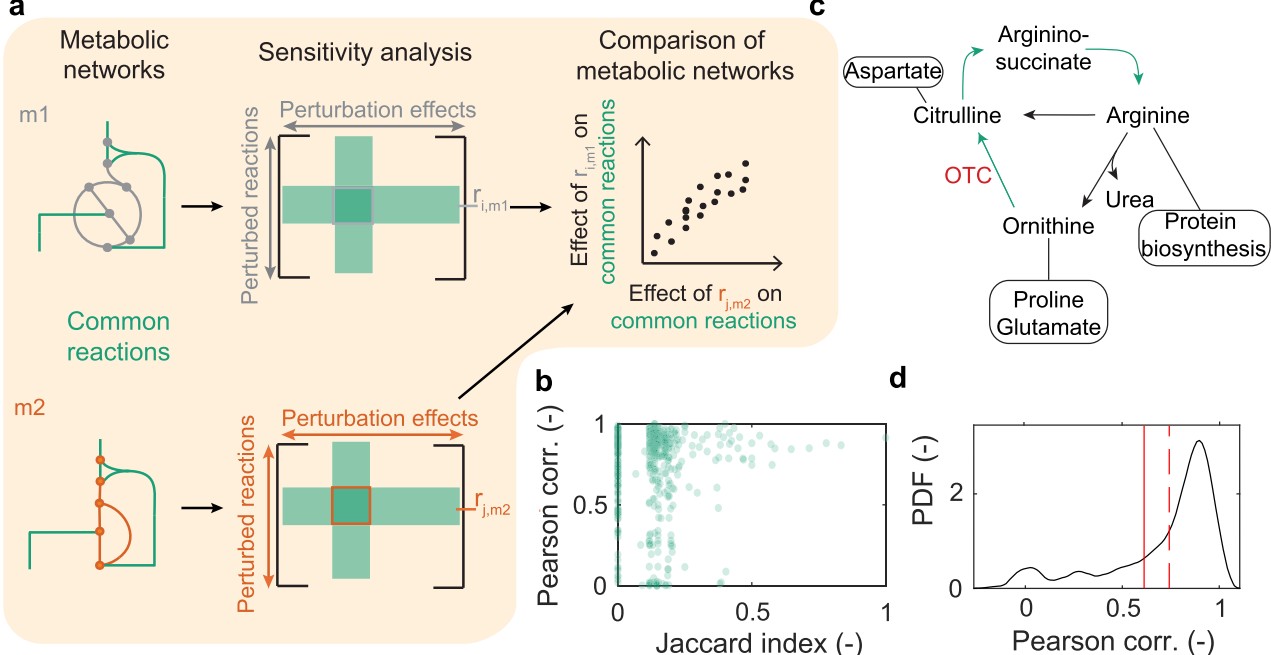

**Fig. 1 | Functional comparison of metabolic networks. a** Example of two metabolic networks with metabolites (nodes) and distinct (grey, orange edges) or common (green edges) reactions. Sensitivities quantify perturbation effects on common reactions. Correlations between each common reaction's effect on network fluxes yield similarity measures (see Methods). **b** Comparison of sensitivity correlations (Pearson) and Jaccard indices computed on sets of reactions belonging to the 1-neighborhood for *E. coli* and *B. subtilis* models; data points: individual reactions. Example for differential effects of perturbing the same reaction (catalyzed by ornithine carbamoyltransferase) in *Bacillus subtilis* and *Escherichia coli*. **c** Circles: adjacent metabolic pathways; red: perturbed reaction, ornithine carbamyl transferase (OTC); green: reactions common to both models; black: *B. subtilis* only. **d** Black line: kernel density estimate of Pearson correlation coefficients of common reactions; red lines: correlations in *E. coli* without (solid) and with (dashed) the two missing *B. subtilis* reactions; PDF: probability density function.

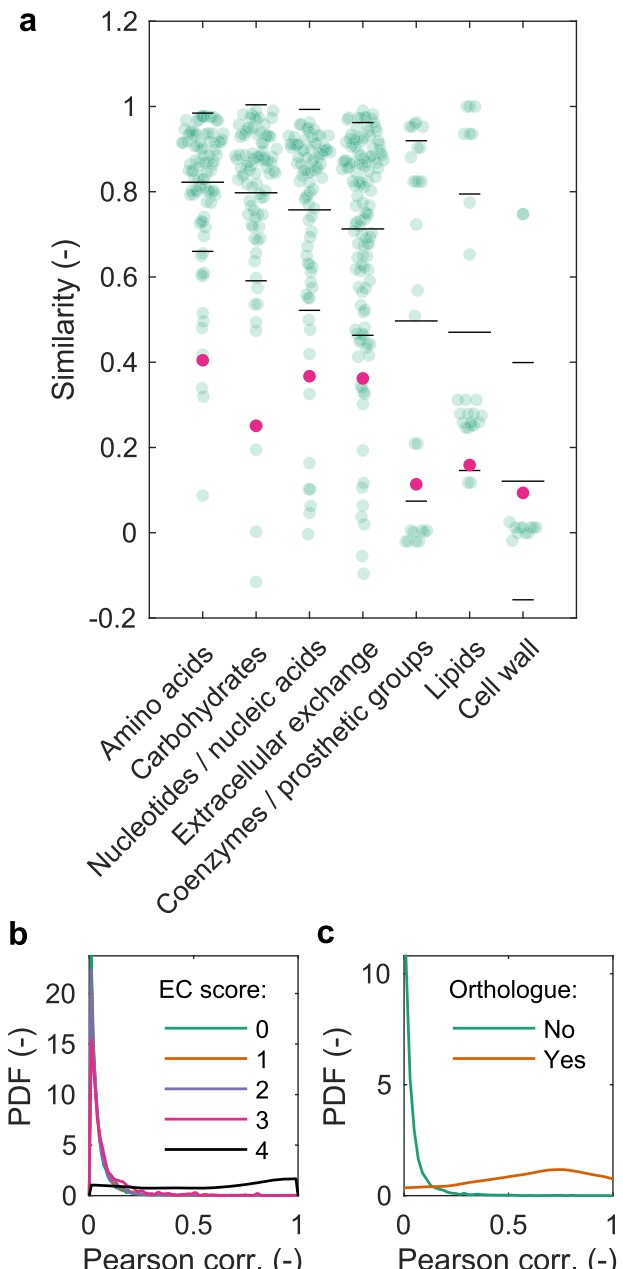

**Fig. 2 | Prediction of subsystem and gene functionality. a** Sensitivity correlations (Pearson) of metabolic reactions (green) and corresponding Jaccard indices (pink) per subsystem (as defined in the GSMs) in the *E. coli* and *B. subtilis* models. Long (short) horizontal black lines: mean (s.d.) per subsystem. Sensitivity similarities reflect gene similarities between human and yeast for Enzyme Commission (EC) numbers (**b**; zero: no similarity; four: maximal similarity; note that curves for scores zero to three overlay and are not distinguishable; for details, see Methods), and gene orthology (**c**; orthologous gene pairs from the OMA database[57]).

## Functional similarity

To assess the biological realism of sensitivity-based predictions, we characterized the functional similarity of metabolic subsystems (sets of reactions with related function) in *E. coli* and *B. subtilis* (Fig. 2a and Methods). Sensitivity correlations indicate that lipid and cell wall metabolism are the least similar, consistent with the bacteria's different Gram status. We also observe a bimodal distribution for reactions in the coenzymes and prosthetic groups subsystem, where the mode with lowest similarity includes mostly reactions in riboflavin metabolism as above. The Jaccard index for each metabolic subsystem gives

similar results (Fig. 2a), but it relies entirely on the subsystem classification of metabolic reactions and cannot reveal fine-grained differences at the reaction level.

To assess the plausibility of reaction-level predictions as well as the potential of comparing biological functions with sensitivity correlations in more complex networks, we next used human and yeast GSMs. We analyzed Enzyme Commission (EC) number similarities between pairs of enzymes, defined as the number of shared levels in the four-level EC number classification. As expected, enzyme pairs with the highest EC number similarities showed higher sensitivity correlations than more unrelated pairs (Fig. 2b). However, enzymes with identical EC numbers do not necessarily have high sensitivity correlations, reflecting their different network contexts. This context-dependence dominates over coarse classification of catalyzed chemical reactions because even a single difference in EC number abolishes correlations. Similarly, when we classified each gene pair of yeast and human as orthologous or not, orthologs had significantly higher correlations (Fig. 2c; one-sided t-test, $P<10^{-10}$, $n=154$ orthologues, $n=1'140'254$ non orthologues). In addition, the correlations span a large range of values, confirming that orthologs are not functionally equivalent[22], despite often catalyzing the same biochemical reaction.

Because Pearson correlations can be unreliable for highly skewed distributions[23] such as here (e.g., Fig. 1d), we also computed copula correlations that are not affected by the underlying marginal distributions (Methods). The two measures can differ for individual reactions, but they are highly correlated (Fig. S1e, $r^2=0.60$, linear correlation) and give identical results for the applications (Figs. 1c, d and 2 vs Fig. S2a–d). Also, reducing the number of reactions to calculate sensitivity correlations had only a small effect (Fig. S2e; albeit more pronounced for Pearson correlation, as expected). Hence, sensitivity correlations establish a detailed, biologically valid, and robust measure of functional similarity.

## Functional alignments and phylogeny

Next, we aimed to align reactions in pairs of GSMs using our measure to evaluate its precision in general, and for only distinctly related metabolic networks. This is possible because our sensitivity-based method yields a one-to-one reaction mapping for each pair of reactions in two networks (Methods). Functional alignment is challenging because even phylogenetically closely-related organisms can have very different metabolic repertoires[24], and structurally similar network parts (e.g., parallel pathways) could have too similar functions to be resolved unambiguously by sensitivity correlations. A previous method for functional network alignment[25] reported 85% correct alignments for 100% common reactions when aligning the yeast GSM iMM904[26] with itself. In contrast, more than 92% of the metabolic reactions were correctly aligned even when using only 1% of the reactions to compute sensitivity correlations (Fig. 3a). Importantly, this indicates that discriminating reaction functions by our measure is insensitive to the number of common reactions, that is, the similarity of metabolic repertoires of two networks.

With alignments being at the basis of phylogenetic analyses, we hypothesized that the sensitivity concept could extend to such comparisons over multiple networks. We define the global similarity of two GSMs as the average sensitivity correlation of all common reactions (Methods). To validate this measure, we compared the yeast model with a randomly reduced version of itself. As expected, both the Pearson and copula correlations decrease as the number of deleted reactions increases (Fig. S3a, b). We then compared 16 manually curated GSMs retrieved from Metanetx[27] that represent 15 species from all kingdoms of life. Consistent with a previous GSM-based analysis of phenotypic evolution[13], average sensitivity correlations decrease with increasing species divergence time and they saturate at high divergence times (Fig. 3b). However, two groups with *B. subtilis* and *Saccharomyces cerevisiae* comparisons suggest higher similarities

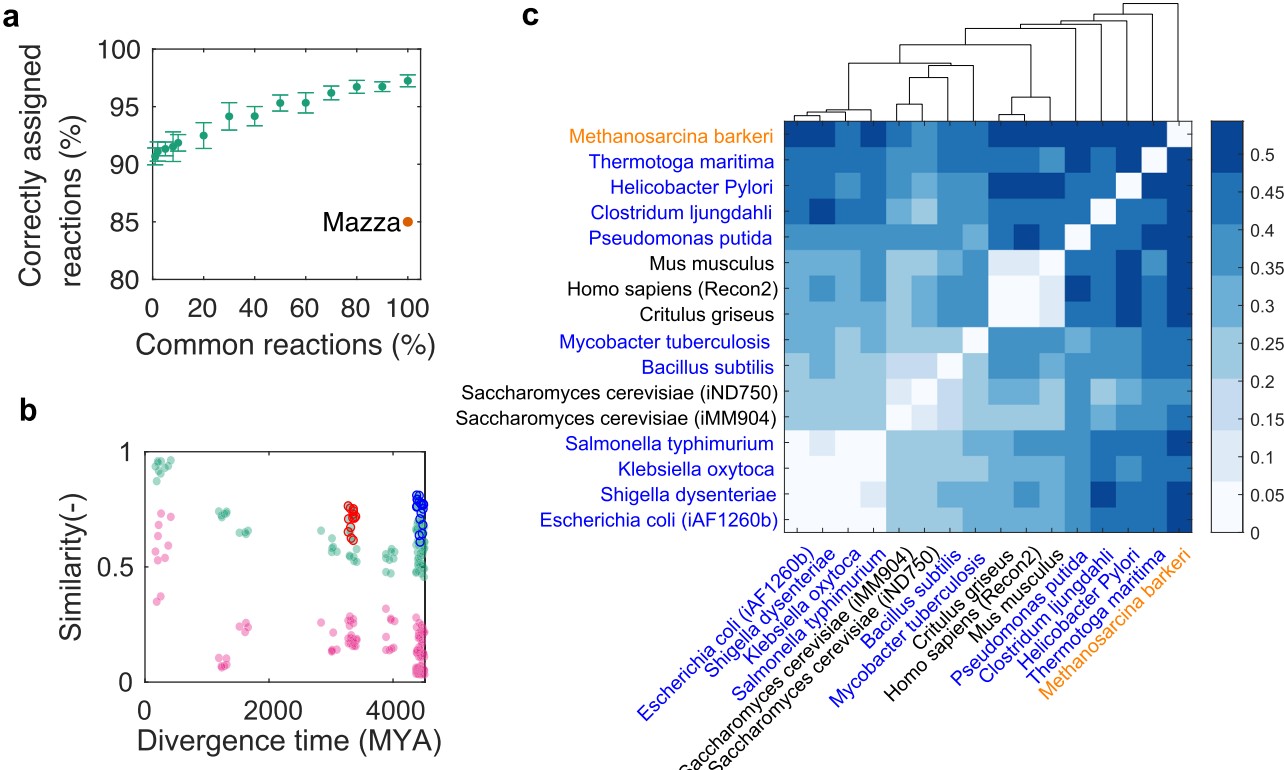

**Fig. 3 | Phylogenetic analysis. a** Alignment of the yeast GSM iMM904 with itself using variable numbers of common reactions. Green: average ± s.d.; red: method by Mazza et al.[25] **b** Average sensitivity correlations (Pearson, green) and Jaccard indices (pink) for model pairs as a function of divergence time. Red circles: comparisons between a subset of the Gram-positive bacteria (*B. subtilis* and *Mycobacterium tuberculosis*) and the Gram-negative bacteria; blue circles: comparisons between yeast and bacteria. **c** Species phylogeny (dendrogram) constructed from the sensitivity dissimilarity matrix (heat map; see Methods) using the unweighted average distance (UPGMA). Archaea: orange; bacteria: blue; eukaryotes: black.

than expected by the general trend. We therefore clustered species hierarchically using the pairwise average sensitivity correlations (Methods).

The resulting species tree (Fig. 3c) is consistent with some aspects of phylogeny (e.g., in separating bacteria, eukaryotes, and the archeon *Methanosarcina barkeri*), but not with others. For example, the metabolically extreme organisms *M. barkeri* (a methanogen) and *Thermotoga maritima* (which does not produce ATP when growing on sulfur) are outliers. Yeast clusters with Gram-positive bacteria (*Mycobacterium tuberculosis* and *B. subtilis*), and not with multicellular eukaryotes. Note that the two yeast models with different network coverage clustered together, indicating a certain robustness to GSM accuracy or completeness. We also confirmed that the inferred species tree is robust to the tree construction method (Fig. S3d) and the correlation measured used (Fig. S3f). In contrast, comparing metabolic repertoires via Jaccard indices (Methods) gives rather binary species distinctions (Fig. S3c). This leads to lower resolution at high divergence times (Fig. 3b) and a qualitatively different inferred tree where, for example, *M. barkeri* clusters with bacteria (Fig. S3e). Hence, in particular distinctions from phylogeny indicate that sensitivities may provide orthogonal information on species-specific metabolic functions and lifestyles.

**Metabolic diversity across bacteria**

To assess the potential orthogonal information in detail, and to exploit the ability to analyze pathways across multiple organisms, we addressed the open question how habitat and taxonomy explain pathway differences. We performed an integrated analysis of metabolic repertoires, functions, lifestyles, and taxonomy, using an established collection of 321 (245 after filtering, see Methods) GSMs for bacteria[13]. The models cover a broad taxonomic and lifestyle (habitat

and physiological) diversity (Fig. S4a) and represent the high diversity of metabolic repertoires in bacteria (Fig. S4b). However, more widespread use of a metabolic reaction across species is not associated with more similar function in the network context as characterized by sensitivity correlations (Fig. S4c), confirming orthogonality of our measure.

We quantify functional similarity across multiple bacteria using normalized biases (z-scores, that is, standard deviations from the mean) to aid interpretation and because of heteroskedacity of sensitivity correlations per reaction (Fig. S4d, e and Methods). Metabolic reactions and subsystems with significant biases respond to perturbations differently than expected from an average ('standard') reaction or subsystem. We classify those with significant positive (negative) normalized bias as 'conserved' ('variable'). This allows a fine-grained analysis of functional conservation across the 245 bacteria, as shown in Fig. 4a for KEGG[28] annotations of subsystems and Pearson correlations. Importantly, this classification is stable over evolutionary distances between species (Fig. S5a–c). It is also robust to alternatives for averaging over reactions (Fig. S5d, e), to significance tests used (Fig. S5f, g), and within SEED[29] subsystem annotations (Fig. S5h, i and Methods).

In this analysis, perturbations in conserved subsystems influence the operation of the entire metabolic networks of different species more similarly than expected. Conversely, we anticipate a conserved subsystem to have limited potential for evolutionary adaptation of its function, for example, because the function is essential for the entire cell, or network constraint enforce a specific function of the subsystem. As one would expect, sensitivity biases identify biomass formation and nucleotide metabolism as conserved (Fig. 4a and S6a, b). Conserved nucleotide metabolism is also predicted by Jaccard indices to assess metabolic repertoire similarity (Fig. S6c) and by comparative genomics[30].

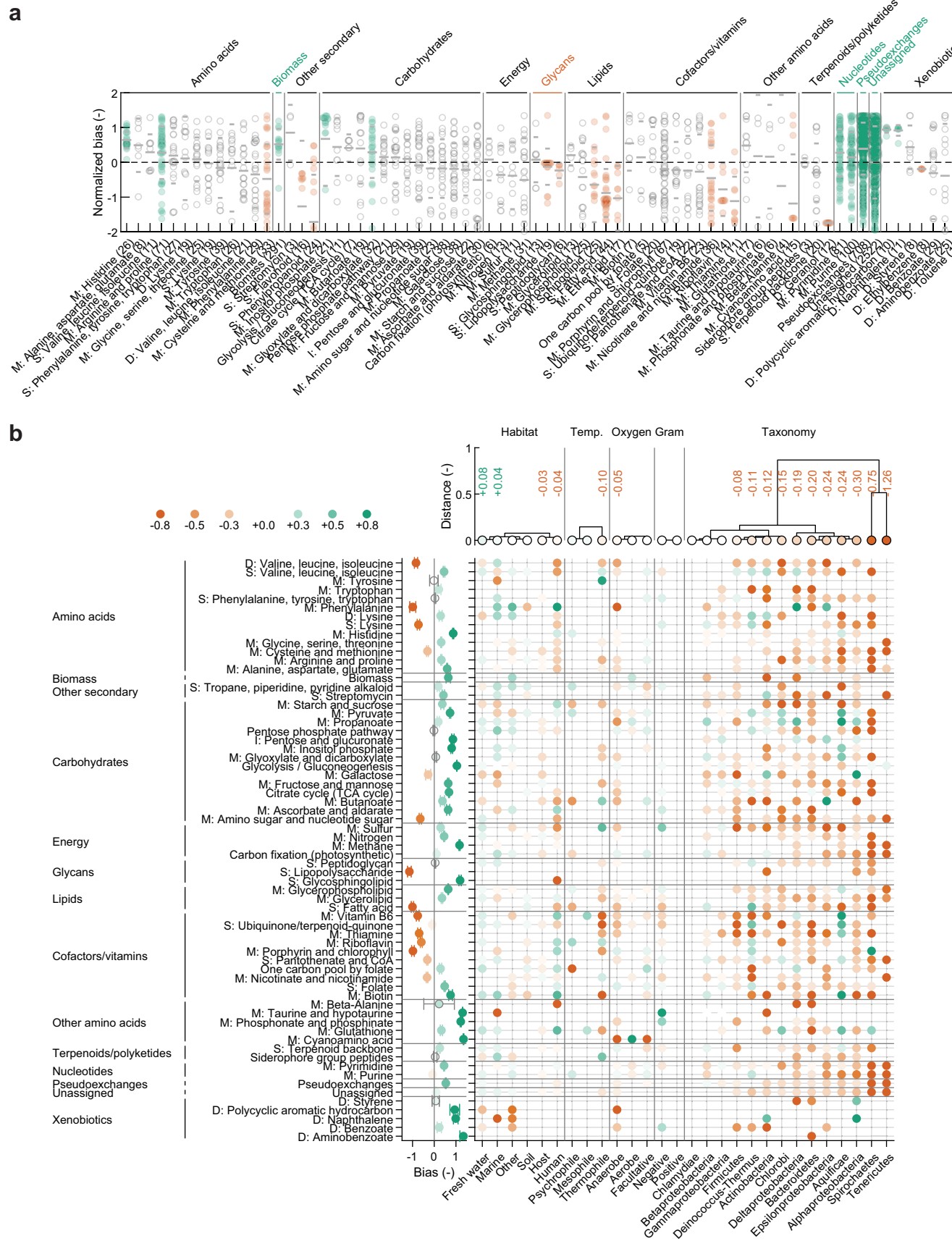

**Fig. 4 | Functional variability of metabolic subsystems in 245 microbial GSMs.** **a** Functional variability of subsystems for KEGG[28] subsystem classification. Points: average normalized bias per reaction (Pearson correlation of sensitivities). Colors show subsystem class: green, conserved; orange, variable, grey: standard (non-significant in two-sample Wilcoxon signed rank test against all reactions, α = 0.05). Horizontal grey lines: subsystem mean (long) and s.d. (short). Coloring of super classes (top) analogously based on subsystem enrichment (hypergeometric distribution, α = 0.05). Abbreviations: M, metabolism; S, synthesis; D, degradation; I, interconversion. Brackets in subsystem labels: number of reactions. Only significant subsystems and subsystems shown in **b** are included. **b** Functional variability depending on NCBI taxonomy, habitat, and physiology (THP) classes (Methods and Supplementary Data 3). Trees (top): Functional distances per THP

class based on mean normalized sensitivity bias over all reactions. Node colors (see color bar) denote regression coefficients for THP variables with respect to normalized sensitivity biases for all reactions; numbers denote values of significant (t-test, α = 0.05) coefficients. Subsystem-specific normalized biases from linear regression (bottom, left): mean ± s.d. over reactions in each subsystem; color-coded effect size, see color bar; grey: not significant. Matrix (bottom, right): Significant regression coefficients for dependence of subsystem mean biases on THP variables (bottom labels). Significance of coefficients: two sided t test, α = 0.05. Significance of regressions: F-test against constant model, α = 0.05. For subsystem analysis, out of the $n = 102$ annotated subsystems, only those with significant regressions are shown. Abbreviations: Temp., temperature.

However, conserved function does not necessarily require a conserved metabolic repertoire. For exchanges with the environment (so-called pseudoexchanges), sensitivities indicate conservation (Fig. 4a) and the metabolic repertoire variability (Fig. S6c). This suggests that, while bacteria have varying exchange repertoires, an exchange's metabolic function is similar in the network context across bacteria. We find the inverse for cofactor and lipid metabolism: despite their conserved metabolic repertoires (Fig. S6c), they are functionally variable (Fig. 4a). Cofactor metabolism is enriched for essential genes[31], but, for example, bacteria use diverse mechanisms for redox balancing[32] and they can evolve new enzyme functions to achieve balance[33]. Functionally variable lipid metabolism may underlie the diversity of bacterial lipids[34], especially when considered together with glycan metabolism for synthesis of the outer membrane.

Such distinctions between repertoire and function become more tangible considering that finer details appear, for example, in amino acid metabolism (Figs. 4a and S6c). We find conservation for histidine, as one would expect from one universal known biosynthesis pathway[35], but functional conservation vs variable repertoire for arginine and proline. There, alternative biosynthesis pathways exist across species, but the functional requirement of providing the amino acids is conserved. We predict especially proline biosynthesis to be functionally conserved (Fig. S8a), consistent with its evolutionary conservation and proline's important role in redox and stress protection[36].

These results on functional conservation agree with previous analyses[37] and copula correlations yield qualitatively very similar functional results (Fig. S7). However, as in prior GSM construction[38] and analysis[39], our analysis depends on manually curated subsystem classifications. Using the SEED classification[29], only some functionally conserved (e.g., biomass, proline) and variable (e.g., cell wall, cofactors) subsystems agree (Fig. S8a), due to limited mapping between KEGG and SEED subsystems (Fig. S8b). For example, having few reactions in SEED nucleotide metabolism prevents reaching significance. This limits biological interpretation, but overall multiple lines of evidence support that sensitivity correlations consistently introduce an orthogonal functional dimension to comparative network analysis.

### Drivers of metabolic diversity
To address how taxonomy and environment influence diversity of metabolic functions across bacterial species, we determined functional conservation or variability induced by single features (e.g., marine habitat). Note that these feature annotations based on widely used databases (see Methods) are incomplete and subject to uncertainties that may affect the analysis. Because taxonomy and environment almost always jointly contribute to predicted functional conservation, we used joint estimation by linear regression to untangle their influences (Methods and Supplementary Data 3). Distances between regression coefficients allowed us to cluster species, for example, by their phyla according to NCBI taxonomy (Fig. 4b). Spirochaetes and Tenericutes are outgroups in the resulting tree, presumably because they are under-represented (three GSMs each) and biased (only host-associated species; Fig. S4a). This tree also

reveals aspects of phylogeny, such as proteobacteria not being monophyletic[40] (Fig. S9a). All trees together show significant impact of both taxonomy and environment on metabolic function, consistent with metabolic diversity within bacterial phyla. With a phylogenetically consistent taxonomy[41,42] (Fig. S9b, c) and other analysis alternatives (Fig. S10), we predict more pronounced taxonomic influences and higher functional similarity within taxa, but environmental factors remain significant for metabolic diversity. When subsampling sensitivity correlations to approximate the effect of inaccuracies in GSMs, we observed only small variations in taxonomic clustering (Fig. S11); the categories discussed next remained stable.

Environmental contributions to metabolic function cluster according to previously defined categories of environmental variability[43], namely host-associated vs. fresh water and marine vs. soil and other (Fig. 4b). However, the estimated regression coefficient for the environments show that environmental variability does not dictate functional variability. We infer metabolic variability for host association, low-variability environments, and conservation for aquatic environments of intermediate environmental variability. In other words, bacteria in watery (host) environments tend to be more functionally similar (dissimilar) than their taxonomic peers. In particular, marine bacteria show a conservative bias, consistent with metabolic niches reducing metabolic variability across taxa[7]. Host environments with comparatively abundant resources and host-specific interactions may afford or even require more metabolic variability of bacteria (e.g., via auxotrophies).

Predicted metabolic functions for bacteria living at low-to-average temperature are conserved, but variable for thermophiles. Functional variability of thermophiles is consistent with previous analyses that showed metabolic networks of thermophiles to be less modular than those of other bacteria[44,45], considering that reduced modularity implies reduced module function[45]. For oxygen requirements, we capture diversity of anaerobic metabolism, and facultative anaerobes as generalists separate. As a negative control, Gram status has no significant influence – it is part of taxon definitions.

Variations of (metabolic) phenotypes according to evolutionary history (represented by taxonomy) and environment are largely unknown and corresponding experimental studies are rare[4,5]. Our approach predicts the experimentally determined impact of taxonomy (high variability in proteobacterial classes and Actinobacteria) vs. environment (low variability in soil) on metabolic phenotypes in soil ecosystems[4] (Fig. 4b). It also suggests more details, namely that β- and γ-proteobacteria are more functionally similar to their peers than α-proteobacteria. As another example, among the dominant classes of functionally conserved psychrophiles (Fig. S4a), we expect higher similarity within γ-proteobacteria than within firmicutes (Fig. 4b). Novel hypotheses such as these are testable in experiments similar to ref. [4].

### Diversity of metabolic subsystems
Estimates of normalized sensitivity biases adjusted for taxonomic and environmental influences (Fig. 4b) overall agree well with those

inferred directly from sensitivity data (Fig. 4a), e.g., regarding variable cofactor and vitamin metabolism. They provide additional evidence for conservation of many metabolic subsystems, but also unexpected findings such as higher conservation of pyrimidine vs purine metabolism. Close connections to amino acid metabolism and cofactor synthesis, respectively, could explain this difference[46].

For habitat and taxonomy influences on subsystems, 755 out of 2856 tested hypotheses were significant (Fig. 4b). These significant hypotheses remained stable under subsampling of sensitivity correlations to a large extent (Fig. S12); this holds for all hypotheses discussed in the following except for the taurine subsystem for which the low number of data points prevented regressions with subsamples. For example, the gut microbiome influences host amino acid and glutathione metabolism in mice[47]; we predict these subsystems as significant for human-associated habitats (Fig. 4b). The most surprising predicted signatures of human-hosted bacteria are: (i) generally strong associations (large absolute biases), indicating high adaptation; (ii) strong conservation of phenylalanine, biotin, and glutathione metabolism, suggesting that adaptation to human requires peculiar functions of these subsystems; and (iii) variability in glycosphingolipid synthesis exclusive to these bacteria, which is intriguing given the underexplored role of bacterial sphingolipids in human immunomodulation and metabolic disorders[48].

To take marine habitats as another example, we predict conserved siderophore metabolism (Fig. 4b). To scavenge scarce iron efficiently in this environment, the release of iron-chelating siderophores appears essential for bacteria[49], explaining functional conservation. The regressions also show higher variability of taurine metabolism (which is conserved on average) in marine environments. This could reflect both the importance of taurine as C- and N-source in general, and the depth-dependence of the availability of taurine and alternative nutrient sources[50].

Finally, we suggest that functional variability facilitates the potentially costly evolution of metabolic cooperation between species. Intriguingly, experimental evolution of mutualism between the γ-proteobacteria *Salmonella enterica* and *E. coli* involved methionine and galactose[51]; those subsystems of γ-proteobacteria are variable among mostly conserved amino acid and carbohydrate metabolism (Fig. 4b). Cooperation via siderophores in marine bacteria, however, is a counter-example: it depends on physico-chemical characteristics of the environment that cannot be captured in GSMs[49]. Our hypotheses could primarily help define relevant metabolic phenotypes for experimental studies of individual microbial species as well as consortia.

## Discussion

We propose sensitivity correlations as a measure to quantify effects of perturbations in metabolic networks to link metabolic repertoires, functions, and their relations to evolutionary history and environment. In contrast to prior work, we do not need to define[15] or randomly sample[14] environments, and we cover exchange fluxes as well as internal fluxes. Combined with our approach's focus of the network context, it thereby enables consistent predictions that were previously inaccessible, for example, on functional conservation of metabolic subsystems. However, uncertainties in GSMs (which could be reduced by probabilistic approaches to network reconstruction[52]), ambiguities in widely used subsystem annotations, and potential biases in the collections of analyzed models limit the biological accuracy of these predictions. For example, detailed reaction-level comparisons of *B. subtilis* and *E. coli* required manual checking. Hypotheses generated by our integrated analysis of bacterial metabolic diversity therefore require empirical validation – and they can simultaneously guide corresponding studies.

We envisage different levels of empirical validation. First, recent advances that increase accuracy and throughput of $^{13}$C metabolic flux analysis[53] can enable systematic testing of sensitivity-based predictions

for individual enzymes, provided targeted (and sufficiently small, e.g., by drug dosing) perturbations can be introduced and a sufficient number of common reaction fluxes can be resolved. For example, one could investigate iso-enzymes predicted to have a most dissimilar effect on network operation in humans and (pathogenic) yeasts as candidates for novel antibiotics. Second, more indirect experiments could be designed that use targeted (e.g., via CRISPR/Cas) or untargeted (e.g., transposon mutagenesis) mutations and indirect readouts such as growth on different nutrient sources for different bacteria to test predictions on subsystem functional conservation with largest effect size. Finally, studies of bacterial ecology in different natural habitats[4] could be designed on our corresponding predictions on drivers of metabolic diversity, for example, focusing on the pronounced predicted differences between β-/γ- and α-/ε-proteobactria.

We consider the concept of sensitivity correlations as the main contribution of this work, and the range of applications presented as proofs-of-principle. Increased species diversity and thereby statistical power could increase taxonomic depth and functional specificity, for example, regarding 'accessory' genomes that enable intra-species metabolic exchanges in microbiomes[54]. Incorporating quantitative characteristics of environments could lead to finer ecological resolution[55]. By allowing such refinements directly, our framework will be instrumental for detailed and systematic studies of relations between metabolic repertoires, phenotypes, evolution, and environment.

## Methods

### Metabolic models and databases

GSMs of yeast (iMM904), human (Recon1), *Bacillus subtilis* (iYO844), and *Escherichia coli* (iJO1366) were retrieved from the BIGG database[56]. We added missing reactions to iYO844, namely a nitric oxide synthase reaction and exchange reactions for nitric oxide. 16 GSMs (Supplementary Data 1) were retrieved from Metanetx[27]. Orthologous pairs of human and yeast genes were from the OMA database[57]. We retrieved 321 automatically reconstructed bacterial GSMs (SEED models) as well as pairwise genetic distances between organisms from ref. [13]. We obtained the reference bacterial phylogenetic tree from ref. [40].

### Curation of SEED models

For every SEED model, we checked that the biomass reaction carries a strictly positive flux when exchange reactions allow uptake of every possible exchanged metabolite. Under this condition, whenever possible, we removed the structurally blocked reactions (reactions that cannot carry any flux) using flux variability analysis (if the minimum and maximum fluxes are both equal to zero, then the reaction is considered blocked)[58]. As a sanity check, we verified that the biomass production before and after the removal of the blocked reactions was identical. When this could not be verified, the original model instead of the reduced model was used (24 models out of 321). The cases where the model reduction did not work are indicated in Supplementary Data 1. Finally, we restricted the model set by two criteria: (i) to include only one representative per species, and (ii) to require a minimum of two species per taxon. The final set then comprised 245 GSMs for the NCBI taxonomy and 242 GSMs for the phylogenetically consistent GTDB taxonomy, as detailed in Supplementary Data 1.

### Annotations for SEED models

We augmented SEED models by taxonomic annotations of bacterial species using the NCBI taxonomy[59] as well as the GTDB taxonomy[42]. Mapping between models and the corresponding databases was performed by species names, including by species synonyms retrieved from the MACADAM database[60]. For habitat and physiology annotations, we used fusionDB[61]. For species with multiple habitat assignments, we automatically identified the main categories ('fresh water', 'marine', 'host' and 'soil'), and subsumed entries that could not be

categorized or of low frequency for SEED models under 'other'. With available evidence (e.g., specific annotations for human gut), we classified 'host' as 'human' more specifically (see Supplementary Data 1). If not already available from fusionDB, we added annotations on Gram status from the Microbe Directory[62,63]. To annotate reactions with subsystems, we first updated the GSMs with current ModelSEED[64] annotations (v2.6.1). To obtain corresponding KEGG[28] subsystem annotations, we relied on reaction aliases provided by ModelSEED (Supplementary Data 2).

## Graph distances
To compute graph distances between pairs of reactions in one GSM, we transformed the GSM to an adjacency reaction graph and applied Dijkstra's algorithm.

## Mathematical notation
We denote the set of GSMs by $M$, and correspondingly the number of GSMs by $|M|$. Denote by $l,m \in M$ two models from this set. Denote by $R$ a set of reactions, by $k \in R$ a reaction in this set, and by $R_l$ the set of reactions in model $l$. Assume that we can identify which two reactions are identical (have identical biochemical formula) in two reaction sets. Then, the set of common reactions of the model pair $(l,m) \in M \times M$ is $R = R_l \cap R_m$. The notation extends to arbitrary subsets of reactions. In particular, we define a metabolic subsystem $j$ as a subset of reactions in any model, $S_j \subseteq \bigcup_{l \in M} R_l$.

We denote subsets of models with respect to the presence of a particular reaction $k$ as $M_k := \{l \in M \mid k \in R_l\}$. This directly leads to the definition of a reaction's (relative) usage across models as:

$$\varrho(k) = \frac{|M_k|}{|M|}.$$

Correspondingly, we define all pairs of models as $P := \{(l,m) \in M \times M, l \neq m\}$ and the subset of model pairs with reaction $k$, $P_k \subseteq P$, as $P_k := \{(l,m) \in M \times M, l \neq m \mid k \in R_l \cap R_m\}$.

## Structural sensitivity analysis
Structural sensitivity analysis[17] quantifies how the perturbation of a reaction flux affects all fluxes in a GSM, assuming minimal total flux adjustment as in the minimization of metabolic adjustment (MOMA) method[18]. Here, we computed absolute structural sensitivities that neither require the definition of a reference flux distribution (as in MOMA[18]), nor the definition of an environment through constraints on exchange fluxes (as in flux variability analysis, FVA[65]). Hence, absolute structural sensitivities characterize network responses independent of a cell's metabolic state or environment.

Specifically, we first characterized each reaction in each GSM using structural sensitivity analysis[17]. For model $l$, we assume that the flux through reaction $k \in R_l$ is perturbed with a disturbance $\delta_k$. The minimal adjustments of fluxes required to return to steady-state, $\mathbf{d}_l$, are obtained by singular value decomposition. It solves the minimization problem:

$$\min_{\mathbf{d}_l} \quad |\mathbf{d}_l|_2$$
$$\text{subject to} \quad \mathbf{N}_l \cdot \mathbf{d}_l = \mathbf{0}$$
$$d_{l,k} = \delta_k,$$

where $\mathbf{N}_l$ is the stoichiometric matrix of the metabolic network and $d_{l,k}$ the element of vector $\mathbf{d}_l$ corresponding to the perturbed reaction $k$.

The sensitivity represents the effect of a perturbation on any reaction relative to the strength of the perturbation. Correspondingly,

we define the vector of sensitivities of each reaction of model $l$ with respect to a perturbation of reaction $k$ as:

$$s(R_l,k,l) = \frac{\mathbf{d}_l}{\delta_k}.$$

This vector has elements $s(i,k,l)$ for $i \in R_l$. Here, we used $\delta_k = 1$.

To make sensitivity computations efficient, GSMs were preprocessed by removing blocked reactions. For reaction pairs in pairs of GSMs, we characterized functional similarity (distance) by correlations (lack of correlations) of absolute structural sensitivities over all common reactions after perturbing a single reaction. For gene-based analysis, we computed sensitivities by simultaneously perturbing all reactions associated with a gene (via the GSM's gene-reaction associations) with the same perturbation magnitude.

## Sensitivity-based correlations and distances
To define the similarity of responses of two models $l,m \in M$ to perturbations in a common reaction $k \in R_l \cap R_m$, we use a correlation function $\rho(\cdot,\cdot)$:

$$S(l,m,k) = \rho(s(R_l \cap R_m,k,l),s(R_l \cap R_m,k,m)),$$

where we sort the two vectors in the same order of reactions. The sensitivity distance between the two models with respect to perturbation of a single reaction is then defined as:

$$D(l,m,k) = 1 - |S_M(l,m,k)|.$$

We extend these concepts from a single reaction $k$ to an arbitrary set of shared reactions $R \subseteq R_l \cap R_m$ for sensitivity correlations as

$$S(l,m,R) = \frac{1}{|R|} \sum_{k \in R} S(l,m,k).$$

This is a quite natural extension because $S(l,m,\{k\}) = S(l,m,k)$.

The average model similarity of two GSMs is then defined as the average sensitivity correlation over all common reactions:

$$S_M(l,m) = S(l,m,R_l \cap R_m).$$

This directly yields an average dissimilarity for pairwise GSM combinations that is interpretable:

$$D(l,m) = 1 - |S_M(l,m)|.$$

## Measures based on reaction sets
The Jaccard index quantifies the similarity of two sets, and we use it to compare the reaction contents of two GSMs $l$ and $m$:

$$J_M(l,m) = \frac{|R_l \cap R_m|}{|R_l \cup R_m|}.$$

To expand this concept to quantify the context similarity of a reaction $k$ in two models, we define a reaction neighborhood via graph distances. Specifically, with $D_G(k,i)$ a function yielding the graph distance between reactions $k$ and $i$, and $\delta$ a distance threshold, the neighborhood $N(l,k) \subseteq R_l$ of reaction $k$ in model $l$ is:

$$N(l,k) := \{i \in R_l \mid D_G(k,i) \leq \delta\}.$$

The Jaccard index for reactions then becomes:

$$J_R(l,m,k) = \frac{|N(l,k) \cap N(m,k)|}{|N(l,k) \cup N(m,k)|}.$$

## Normalized similarities

To account for potential biases due to different sets of common reactions in GSM pairs, we compute normalized similarities as

$$\tilde{S}(l,m,k) = \frac{1}{\sigma}(S(l,m,k) - \hat{b}_0 - \hat{b}_1 \cdot \varrho(k)).$$

Here, $\hat{b}_0$ and $\hat{b}_1$ are estimated intercept and slope of a linear regression of the average reaction similarity

$$S_R(k) = \frac{1}{|P_k|} \sum_{(l,m) \in P_k} S(l,m,k)$$

as a function of reaction usage $\varrho(k)$ jointly for all $k$, and $\sigma$ is the root mean squared error of the linear regression. Without significant correlations between reaction usages and sensitivity correlations in our data (Fig. S4c), these definitions amount to computing z-scores.

We expand this normalization to more general similarities by replacing $S$ with $\tilde{S}$ appropriately, leading to average normalized similarities for pairs of models, $\tilde{S}_M(l,m)$, and average normalized reaction similarities, $\tilde{S}_R(k)$. For conciseness, we term the latter 'normalized bias'.

## Subsystem analysis

To study variability of metabolic subsystems across all models, we partition the set of unique reactions into disjoint subsets $S_j$ according to subsystem classification and include an additional subset for reactions with unassigned subsystem.

For sensitivity-based analysis, the average normalized bias $\tilde{S}_S(j)$ of a subsystem $j$ is defined as:

$$\tilde{S}_S(j) = \frac{1}{|S_j|} \sum_{k \in S_j} \tilde{S}_R(k).$$

The classification of subsystems into categories relied on three alternative approaches:

(i)   Subsystem mean by reaction: Distribution of $\tilde{S}_R(k)$ for all $k \in S_j$.
(ii)  Subsystem mean, alignments: Distribution of $n_s = 100$ samples from $\tilde{S}(l,m,k)$ for any $(l,m) \in P_k$ and any $k \in S_j$.
(iii) Subsystem mean, bootstrap: Distribution of $n_r = 100$ estimated averages of $n_s = 100$ samples from $\tilde{S}(l,m,k)$ for any $(l,m) \in P_k$ and any $k \in S_j$.

For analyses based on reaction sets, we define a corresponding measure to compare subsystems $S_j$ as:

$$J_S(l,m,S_j) = \frac{|S_j \cap (R_l \cap R_m)|}{|S_j \cap (R_l \cup R_m)|}.$$

For a subsystem classification that is consistent with the sensitivity-based approach, we compute normalized Jaccard indices $\tilde{J}_S(l,m,S_j)$ by subtracting the average of $J_M(l,m)$ over $P$ and dividing by the corresponding standard deviation. We evaluate the distribution of $n_s = 100$ samples from $\tilde{J}_S(l,m,S_j)$.

## Enzyme similarity

To validate sensitivity-based predictions with independent measures, we relied on Enzyme Commission (EC) numbers. They provide a hierarchical numerical classification for enzyme functions composed of four levels (numbers) that represent a progressively finer classification. To compare two enzymes, we define an EC similarity level according to the maximal level up to which their EC numbers coincide. This leads to five levels of similarity between zero (no similarity) and four (completely identical EC numbers). When several EC numbers were mapped to the same enzyme, we used the maximal similarity level of all possible pairwise comparisons. We proceed identically when several EC numbers were mapped to one reaction through its catalyzing enzymes.

To globally characterize the aligned reactions in two GSMs $l$ and $m$ via their EC number similarity, we define an EC score akin to the Kullback-Leibler divergence:

$$E(l,m) = \sum_{i=0}^{4} a_i \cdot \log\left(\frac{a_i}{n_i}\right),$$

where $a_i$ is the fraction of aligned reactions $R \subseteq R_l \cap R_m$ with an EC number similarity level $i$, and $n_i$ is the fraction of reaction pairs with EC number similarity $i$ in the null model (all possible reaction pairs between two models).

## Alignment

Sensitivity distances between all possible pairs of reactions characterize all possible mappings between reactions in two GSMs. The assignment problem corresponds to identifying the best reaction mapping using solely sensitivity distances, with blinded reaction identities. We solved it using the Jonker-Volgenant algorithm[66], which selects the set of pairs of reactions with total minimum sensitivity distance. Compared to ref. [25], our method is faster (30 min versus 48 h per alignment) because it requires only one optimization per reaction. It returns a set of mapped reactions with their individual sensitivity distances, and a set of unmapped reactions in the larger GSM. For validations, we aligned the yeast GSM with a copy of itself, after randomizing the order of reactions in the copy and varying the number of common reactions. We measured performance by the number of correctly aligned reactions.

## Tree construction

We characterized the pairwise distance between GSMs by sensitivity-based average dissimilarity as well as Jaccard index distances. Trees for analyses of habitat and taxonomy used estimated regression coefficients (see below). We applied the Unweighted Pair-Group Method with Arithmetic mean (UPGMA) for all tree constructions. The phylogenetic tree based on divergence times was retrieved from TimeTree[67].

## Taxonomy, habitat, and physiology analysis

We analyze metabolic flux phenotypes with respect to five classes of features (habitat, temperature preference, gram status, oxygen preference, and phylum). The features in each class can be mutually exclusive (e.g., gram positive or gram negative status), or not (e.g., a microbe may have more than one preferential habitat). Importantly, the feature classes are not independent (e.g., often, taxonomy definitions are based on the bacteria's gram status). Therefore, all features have to be accounted for simultaneously in the analysis.

We encode features in row feature vectors $\mathbf{F}$, where $\mathbf{F}_l$ is the feature vector for model $l$. The value of element $p$ of $\mathbf{F}_l$, $F_{l,p}$, denotes if an organism has feature $p$ (1) or not (0), or if it is undefined (−1). We restrict the analysis to those pairs of models where both models have identical and completely specified feature vectors, denoted as $P_F \subseteq P$. These model pairs are defined as:

$$P_F := \{(l,m) \in M \times M, l \neq m \mid \forall p : F_{l,p} = F_{m,p}, F_{l,p} \in \{0,1\}\}.$$

To quantify how similarities depend on features, we estimate linear models of the form $\mathbf{Y} = \mathbf{X} \cdot \mathbf{b} + \varepsilon$. We construct the design matrix $\mathbf{X}$ with one row $[1\,\mathbf{F}_l]$ for each $(l,m) \in P_F$.

To assess the impact of features on average normalized model similarity, the elements of the response matrix $\mathbf{Y}$ are $\tilde{S}(l,m,R_l \cap R_m)$ for $(l,m) \in P_F$. The resulting coefficient estimates $\hat{b}_p$ are used for tree construction. Correspondingly, for the analysis of a specific subsystem $j$, the elements of $\mathbf{Y}$ are $\tilde{S}(l,m,S_j \cap R_l \cap R_m)$ for $(l,m) \in P_F$.

## Statistical analysis

To assess significance of normalized biases for subsystem classification, two tests were used as indicated in text and figures: (i) two-sided Wilcoxon signed rank test with $\alpha = 0.05$, and (ii) empirical p-value determination after 10'000 repeats of sampling reactions within a given subsystem and all reactions with replacement according to the number of reactions per subsystem and computing the average sample differences. Enrichment of conserved or variable subsystems in their parent classes was determined by one-sided tests via the hypergeometric cumulative distribution, again with confidence level $\alpha = 0.05$.

For taxonomy, habitat, and physiology analysis, we evaluated the impact of features on subsystem variability via significant coefficient estimates of significant linear regressions. Significance of linear regressions was determined by F-tests against the constant null model without correction for multiple testing ($\alpha = 0.05$). P-values for tests that coefficients are zero were based on the t-statistic (two-sided) and we used $\alpha = 0.05$ for significance.

## Implementation

All calculations were performed with Matlab 2019b, node version (Mathworks, Natick / MA) and Gurobi Optimizer (version 8.1.1).

## Reporting summary

Further information on research design is available in the Nature Portfolio Reporting Summary linked to this article.

## Data availability

The datasets generated during and/or analyzed are available at https://doi.org/10.3929/ethz-b-000598615. All the data and code for reproducing figures in the main text and the supplementary information are provided in the GitHub repository (see Code availability). We used the following publicly available datasets: 1. Orthologues: retrieved using the OMA database, available at https://omabrowser.org/cgi-bin/gateway.pl?f=PairwiseOrthologs&p1=HUMAN&p2=YEAST&p3=EntrezGene 2. iMM904, iJO844, iJO1366, and Recon1 models were downloaded from Bigg (http://bigg.ucsd.edu/). 3. MetaNetX models were retrieved from MetaNetX (https://www.metanetx.org/). 4. SEED models were downloaded from the publication: Plata, G., Henry, C. & Vitkup, D. Long-term phenotypic evolution of bacteria. Nature 517, 369–372 (2015) (http://vitkuplab.c2b2.columbia.edu/phenotypes/) 5. NCBI taxonomy, available at https://www.ncbi.nlm.nih.gov/taxonomy 6. GTDB taxonomy, available at https://data.gtdb.ecogenomic.org/releases/latest/ 7. Species synonyms (MACADAM database), available at http://macadam.toulouse.inra.fr/doc/MACADAMDatabase.zip 8. Habitat and physiology annotation (FusionDB), available at https://services.bromberglab.org/fusiondb/explore 9. Gram status (Microbe directory), available at https://github.com/microbe-directory/microbe-directory/blob/master/data/microbe-directory.csv. 10. Model SEED annotations and reaction aliases, available at https://github.com/ModelSEED/ModelSEEDDatabase/blob/master/Biochemistry/

## Code availability

Custom code for the analysis is available at https://doi.org/10.3929/ethz-b-000598615 and the GitHub repository https://gitlab.com/csb.ethz/functionalcomparisonmetabnetworks/-/tree/main.

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

## Acknowledgements
We thank Hans-Michael Kaltenbach, Sean Froese and Uwe Sauer for discussions and comments.

## Author contributions
C.R. and J.S. conceived the study. C.R. performed computations. C.R. and J.S. performed the analysis and wrote the manuscript.

## Competing interests
The authors declare no competing interests.
