## [Peer Review File · Nature Communications]

Functional comparison of metabolic networks across speciesReviewer #1 (Remarks to the Author):

This manuscript describes an interesting approach for the comparison of the metabolic functions of different organisms, and its application to understanding the breadth and diversity across microbial taxa. The method cleverly captures the network context of metabolic reaction perturbations, using efficient calculations from stoichiometric modeling. The manuscript is overall well written and inspiring, and the methods are thorough and rigorous. However, I have some concerns and questions that should be addressed before this article is considered for publication.

1. My major concern has to do with the dependence of flux perturbations on the choice of environmental conditions, which does not seem to have been addressed at all in this work. One key question is how the responses to perturbations, and correlations between different GSMs, depend on environmental conditions (both the choice of carbon, nitrogen sources etc., and the specific values of upper bound for import fluxes, which determine the limiting nutrient). What environment was used for the current calculations, and why? How would calculations performed under different environments affect the results of this work? Could it be that two models are very similar under one environmental condition, but substantially different under another condition?

2. The correlation between the sensitivity to a given reaction in two GSMs will depend on the number of reactions in common between the two models. To make a concrete example, if models A and B have 10 reactions in common, while models B and C have 100 reactions in common, the correlation between perturbed fluxes in A and B may behave differently than the correlation between B and C (in terms of statistical significance, sensitivity to noise, etc.). Is this a potential issue? It would be helpful to clarify.

3. Line 85: The authors start a paragraph with "To validate reaction-level predictions..." I am not sure I would call it a validation. I would think of validation as something involving experimental measurements. It would actually be interesting to obtain some idea of whether the perturbation method as a measure of similarity across corresponding reactions would truly display similar patterns if based on experimental measurements. I know this would be very difficult to do, but the authors may want to comment on this in the discussion. (see also comment below)

4. I must admit that I am bit confused about the sensitivity-based alignment. I think I understand the idea, but what I do not understand is why, upon trying to align a model to itself, it is challenging to do a perfect alignment, i.e. find pairs of reactions with identical flux perturbation patterns. Perhaps it would be helpful to improve the explanation and add more details in the Methods or Supplementary Methods sections.

5. At Line 136 the authors write: "Hence, in particular distinctions from phylogeny indicate that sensitivities may provide orthogonal information on species-specific metabolic functions and lifestyles." Could the authors have more specific information, e.g. on what kind of biological information is captured by the sensitivity-based phylogeny, relative to gene-content based phylogeny?

6. The analysis of functional variability and correlation with environment and taxonomy for the 245 GSMs is very interesting. Here, again, however, the issue of dependency on environmental conditions used in the metabolic flux models worries me. How different would the trees and regression maps look if you chose a different set of environmentally available molecules (e.g. anaerobic deep sea conditions, vs. human gut conditions?). I understand that it may not be viable to repeat all analyses under an array of different environmental conditions, but it would be good to know how sensitive the results might be, and perhaps consider choosing a few key environment on which to test robustness for key conclusions.

7. GSMs from different repositories may have different degrees of accuracy and completeness, based also on the amount of manual curation and refinement. Given that some of the analyses are performed on different sets of models from different sources, do the authors have any expectation or comments on how these differences may affect their results?

8. In the discussion the authors mention that "Hypotheses require empirical validation": It would be useful for the authors to be more specific on what kind of experimental follow up they would envisage as ideal for testing specific hypotheses, and how an experimentalist could prioritize specific hypotheses to test.

Minor comment:

Fig. 2b: it is a bit tricky to resolve curves and colors. I understand that many curves may be overlapping, with little that can be done to improve readability. But it should be pointed out in the legend which curves are overlapping. Also, two curves have very similar green color, which may be a source of ambiguity.

Reviewer #2 (Remarks to the Author):

The manuscript by Ramon and Stelling summarises the results of inter-species comparisons between metabolic networks based on a stoichiometric matrix-based metric (previously published by the last author). The metric assesses similarities in the sensitivities of the networks in question in response to a perturbation at shared reaction nodes. Compared to commonly used metabolism comparison metrics (e.g., Jaccard distance or EC mapping similarities), the metric used here accounts for the structural dependencies between reactions within a network. Thus, the comparison is much more meaningful and hints at structural conservation (or lack thereof) of the networks in question. While I generally find the approach very useful, there are several points where the study and the manuscript is unclear / requires improvement.

1. The use of 'function' and 'functional' in the manuscript is very confusing and often inaccurate. I think 'Structural comparison', for example, would be a much more accurate description. The function is generally, and rightfully, thought of as a phenotype or contribution to a, say, biochemical pathway. Perturbation sensitivity of a reaction does not inform much about the function of that reaction or organism per se; two networks could be very different their sensitivity profiles but could be functionally similar under a set of environments (consider two organisms with each having a different set of parallel pathways).
2. The method section on sensitivity calculation is very incomplete. While I appreciate that the method is published previously, there are several details that are important: in particular, under what media conditions were the sensitivity calculated? It would also be important to put the method in the context of other widely used GSM-based metrics such as flux variability analysis.
3. Figure 2. (Note that there is no panel d as the caption mentions.) The panel (b) seems like a circular argumentation, EC mappings are used in GSM reconstruction, so it is expected that sensitivity similarities would reflect gene similarities.
4. The overall message/conclusion is unclear: are the authors proposing a new metric for pathway/subsystem identification? The conclusion that the framework can help in joint interpretation of traits, evolution and environment is quite vague, especially with a fuzzy definition of trait (similar discussion as above for function).

Reviewer #3 (Remarks to the Author):

Ramon and Stelling introduce the application of structural sensitivity analysis as a way to compare the function of metabolic networks across organisms. Instead of comparing the composition of metabolic networks across species or values of specific traits inferred from corresponding models, the authors focus on the similarity of responses to perturbations of individual reactions between organisms. They use this framework to, among other proof-of-principle applications, assess the contribution of taxonomy and the environment to the functional differences inferred by their

method. I believe the approach presented is novel and potentially useful. The methods used are mostly appropriate and support the main claims of the manuscript. The main points to revise relate to the clarity in the interpretation of the functional conservation/variability examples, and the robustness of the metric to metabolic model completeness.

Importantly, the comparison of metabolic phenotypes inferred from metabolic reconstructions with regards to taxonomic or environmental classifications is not a novel concept. For example, Bernstein et al (<https://elifesciences.org/articles/39733>) predicted the biosynthetic capabilities of 456 oral microbes and, similar to the present study, defined a functional distance between microorganisms that was associated with taxonomy and co-occurrence data. Zarecki et al. (<https://dx.plos.org/10.1371/journal.pcbi.1003726>) used the metabolic reconstructions of 2,529 microorganisms to predict minimum growth requirements for growth and compared those predictions with corresponding ecological and genomic traits. Therefore, the novelty of the current study is mainly about 1) the nature of the measure of functional similarity proposed and 2) the focus on the relative contribution of evolutionary history versus environment to the variability of said measure.

Specific comments

1. Line 13. In the abstract the sentence "model-predicted traits were only employed at the species level" should be revised in light of the above references. Similarly, for Line 38, although reference 6 did not make links to genotype or environment, the above papers that did should be cited.

2. Line 58. For the comparison between sensitivities and Jaccard similarities, the size of neighborhoods used to obtain the results in Fig 1b is not specified. This threshold is only defined as δ in the supplementary methods. Thus, it is not clear whether the size of these neighborhoods affects the correlation between the two measures.

3. Fig 1d. I believe the description of the solid and dashed red lines in the figure legend are inverted. Please check.

4. Line 85. The analysis shows that reactions associated with orthologous genes or reactions with the same EC number have higher sensitivity correlations than non-orthologous reactions and reactions with different EC numbers. This suggests that identical reactions are not always functionally equivalent across species, but does not necessarily validate reaction-level predictions, as the authors claim at the start of the paragraph. This is because reactions with EC scores 0-3 as well as non-orthologs all have the same low sensitivity correlation. Instead, the authors should look at the sensitivity correlations of reactions within the same pathway, or reactions at increasing distances in the network graph. As the main point would be that functionally related reactions have higher sensitivity correlations.

5. Line 101. For clarity, it should be specified that the number of common reactions refers to the reactions used to calculate the sensitivity correlations as opposed to the number of common reactions in the models used to calculate the sensitivities.

6. Line 112. Same as above, it should be made clear that common reactions refers to the number of reactions used to calculate the correlations. Not common reactions in the models.

7. Line 148. Why are normalized biases calculated relative to the regression line between reaction usage and sensitivity correlations (as described in the supplementary methods) if there is no correlation between those two variables according to Figure S4c? Would the same results be obtained if a single mean and standard deviation across all reactions was used for normalization?

8. Lines 165-176. Though these are interesting observations, it is unclear how the authors are interpreting processes classified as conserved or variable. For instance, the fact that there's only one universal histidine biosynthesis pathway does not necessarily imply that such pathway is functionally conserved based on sensitivity correlations. Specifically, the result also indicates that the reactions most affected by fluxes through the histidine pathway are similar across species.

Perhaps the authors can highlight with specific examples that the sensitivity results are not always explained by the universality of the pathways but by the processes they influence across the whole network.

9. Line 202. Figure 4b. The normalized biases calculated for the nodes of the environment trees are very small (close to 0), so the color scheme does not easily allow seeing the patterns described in the text. I suggest adding a different representation, or supplementary figure with the mean biases for each of the environmental categories.

10. Line 202. In general, the results in this paragraph are hard to understand. The authors start by saying that host environments associate with low functional variability compared to water and soil, but then say the opposite result is true after controlling for other factors? which factors? where are these results shown in the figures? In the same paragraph, the authors say that bacteria in low-temperature habitats are functionally conserved compared to thermophiles and that this is consistent with reference 41 that found the metabolic networks of bacteria living at high temperatures to be less modular and less flexible. How is this lack of flexibility consistent with more functional variability? Shouldn't the opposite be true?

11. Line 240. The authors say that conserved siderophore metabolism could enable cooperation, but then hypothesize that functional variability facilitates the evolution of cooperation. For this and the point above, the authors should devote some space to explain how they are interpreting their results for the examples provided and not just point to possibly relevant publications, as these connections are not obvious.

12. Line 254. It is important to provide an estimate of the impact of model errors on the predictions by the proposed method. Would the obtained results change significantly for different metabolic models for the same species (e. g. generated via different methods)? Based on the results in figure S3b, would a similar strategy: deleting different numbers of random reactions from selected models, indicate the robustness of the sensitivity correlation metric towards model ambiguities or incompleteness? I think the authors can easily comment and quantify this by applying that strategy to other models.

13. Line 282. It would help for the authors to specify that the sensitivities calculated are absolute sensitivities, and not relative sensitivities, such that the analysis does not require that reference fluxes or simulation conditions are specified. A sentence early in the manuscript clarifying this would facilitate understanding of the methods and interpretation of the results.

Hoping that this helps more clearly communicate this research,
Germán Plata

RESPONSE TO REVIEWER COMMENTS

Reviewer #1 (Remarks to the Author):

This manuscript describes an interesting approach for the comparison of the metabolic functions of different organisms, and its application to understanding the breadth and diversity across microbial taxa. The method cleverly captures the network context of metabolic reaction perturbations, using efficient calculations from stoichiometric modeling. The manuscript is overall well written and inspiring, and the methods are thorough and rigorous. However, I have some concerns and questions that should be addressed before this article is considered for publication.

Thank you very much for the encouraging, positive comments.

1. My major concern has to do with the dependence of flux perturbations on the choice of environmental conditions, which does not seem to have been addressed at all in this work. One key question is how the responses to perturbations, and correlations between different GSMs, depend on environmental conditions (both the choice of carbon, nitrogen sources etc., and the specific values of upper bound for import fluxes, which determine the limiting nutrient). What environment was used for the current calculations, and why? How would calculations performed under different environments affect the results of this work? Could it be that two models are very similar under one environmental condition, but substantially different under another condition?

We apologize for the confusion caused by us not completely specifying the methods for sensitivity analysis. Here, we used absolute sensitivities, which do not rely on reference flux distributions or constraints defining the environment. Consequently, we compare network responses in a ‘generic’ environment and, for example, cannot compare different environmental conditions with this approach. We have now included the following clarification at the start of the Results section (l. 54; line numbers refer to the manuscript version with markup): ‘Here, we computed absolute sensitivities, which do not assume a specific operating state of the network or a specific environment¹⁷. These sensitivities are based on infinitesimal perturbations, which are too small to be affected by specific constraints such as the availability of nutrients in a specific environment.’ In addition, we have expanded the Methods section on sensitivity analysis (see also response to reviewer #2, point 2).

2. The correlation between the sensitivity to a given reaction in two GSMs will depend on the number of reactions in common between the two models. To make a concrete example, if models A and B have 10 reactions in common, while models B and C have 100 reactions in common, the correlation between perturbed fluxes in A and B may behave differently than the correlation between B and C (in terms of statistical significance, sensitivity to noise, etc.). Is this a potential issue? It would be helpful to clarify.

We analyzed the effect of varying numbers of common reactions in two contexts:

(i) For the *E. coli* and *B. subtilis* models, we used bootstrapping to estimate the effect on sensitivity correlations. The data in Fig. S2e shows that with reduced sets of common reactions, the mean sensitivity correlations remain stable (in the range investigated), but the variance of the estimates increases. From this, we concluded in l.115 that ‘... reducing the number of reactions to calculate sensitivity correlations only had a small effect ...’

(ii) The functional alignment of the yeast network with itself (Fig. 3a) demonstrates that with respect to discriminating between reactions according to their responses to perturbations, uncertain estimates because of low numbers of common reactions (in this case, we reduced the fraction of common reactions by up to 100-fold, down to 10's of common reactions) also had only small effects (~10% reduction in alignment accuracy). We now clarify the implication of this analysis for the potential issue of varying common reactions between model pairs; please see also our response to point 4 below.

3. Line 85: The authors start a paragraph with “To validate reaction-level predictions...” I am not sure I would call it a validation. I would think of validation as something involving experimental measurements. It would actually be interesting to obtain some idea of whether the perturbation method as a measure of similarity across corresponding reactions would truly display similar patterns if based on experimental measurements. I know this would be very difficult to do, but the authors may want to comment on this in the discussion. (see also comment below)

We appreciate the reviewer's point. First, we have re-formulated the start of the paragraph to “To assess the plausibility of reaction-level predictions as well as the potential of comparing biological functions with sensitivity correlations in more complex networks, we next used human and yeast GSMs.” Second, a new paragraph in the Discussion now outlines the rationale for validation with experimental measurements, and gives specific examples; for the paragraph cited by the reviewer (please also see below for comments by reviewer 3), we added the following:

“... recent advances that increase accuracy and throughput of ^{13}C metabolic flux analysis can enable systematic testing of sensitivity-based predictions for individual enzymes, provided targeted (and sufficiently small, e.g., by drug dosing) perturbations can be introduced and a sufficient number of common reaction fluxes can be resolved. For example, one could investigate iso-enzymes predicted to have a most dissimilar effect on network operation in humans and (pathogenic) yeasts as candidates for novel antibiotics.”

4. I must admit that I am bit confused about the sensitivity-based alignment. I think I understand the idea, but what I do not understand is why, upon trying to align a model to itself, it is challenging do a perfect alignment, i.e. find pairs of reactions with identical flux perturbation patterns. Perhaps it would be helpful to improve the explanation and add more details in the Methods or Supplementary Methods sections.

We have completely revised the corresponding paragraph to clarify the challenges of functional alignment (e.g., when parallel pathways with very similar function are considered), and to explain in more detail why we considered the alignment problem (to assess the discriminatory power of sensitivity correlations, and to evaluate the effect of network distances in terms of common reactions; please see also point 2 above). In the Methods section, we added the following for clarification: “The assignment problem corresponds to identifying the best reaction mapping using solely sensitivity distances, with blinded reaction identities.”

5. At Line 136 the authors write: “Hence, in particular distinctions from phylogeny indicate that sensitivities may provide orthogonal information on species-specific metabolic functions and lifestyles.” Could the authors have more specific information, e.g. on what kind of biological information is captured by the sensitivity-based phylogeny, relative to gene-content based phylogeny?

We aim to capture the additional biological information by decomposing the variance of sensitivity correlations based on taxonomy (related to phylogeny, but not identical to it), physiology, and habitat and demonstrate that the correlations capture all three categories of factors, in contrast to the corresponding results for metabolic reaction content alone as characterized by Jaccard indices. To make the transition to the subsequent Results sections dealing with this analysis clearer, we have revised the text (l.162) as follows: “To assess the potential orthogonal information in detail, and to exploit the ability to analyze pathways across multiple organisms, we addressed the open question how habitat and taxonomy explain pathway differences.”

6. The analysis of functional variability and correlation with environment and taxonomy for the 245 GSMs is very interesting. Here, again, however, the issue of dependency on environmental conditions used in the metabolic flux models worries me. How different would the trees and regression maps look if you chose a different set of environmentally available molecules (e.g. anaerobic deep sea conditions, vs. human gut conditions?). I understand that it may not be viable to repeat all analyses under an array of different environmental conditions, but it would be good to know how sensitive the results might be, and perhaps consider choosing a few key environment on which to test robustness for key conclusions.

Thank you for the encouraging comment. As stated above (point 1), environmental conditions did not enter the computation of sensitivity correlations as inputs in any form. The trees and regression maps depend on the classification of organisms (e.g., regarding their preferred environment), for which we used standard databases. In this regard, unknown, and therefore hard-to-model uncertainties exist, which we now acknowledge specifically (l.225) by “Note that these feature annotations based on widely used databases (see Supplementary Methods) are incomplete and subject to uncertainties that may affect the analysis.”

7. GSMs from different repositories may have different degrees of accuracy and completeness, based also on the amount of manual curation and refinement. Given that some of the analyses are performed on different sets of models from different sources, do the authors have any expectation or comments on how these differences may affect their results?

With respect to one effect of variations between GSMs, the number of common reactions (which could vary due different degrees of model completeness or annotation), we refer to point 2 above – our analyses showed only minor effects on the reported results.

In the phylogenetic analysis (Fig. 3c), we included two models for *S. cerevisiae* (iND750 and iMM904) with substantially different coverage of the metabolic network, and these models clustered together, indicating that – at this level of analysis – differences in model construction did not affect the conclusions. We have added a corresponding statement at l.154: ‘Note that the two yeast models with different network coverage clustered together, indicating a certain robustness to GSM accuracy or completeness.’

Different GSM origins or construction methods, however, clearly affect results regarding fine-grained predictions such as reaction-level predictions in the *E. coli* vs *Bacillus* comparison, which needed to be checked manually. We now state this in the Discussion explicitly (l.319).

For the reasons pointed out by the reviewer, we performed the large-scale analyses of microbial metabolic networks on automatically reconstructed models, where one can assume that models are less subject to source bias. The revision includes a detailed robustness analysis of these predictions – please see our response to reviewer #3, point 12 for details.

8. In the discussion the authors mention that “Hypotheses require empirical validation”: It would be useful for the authors to be more specific on what kind of experimental follow up they would envisage as ideal for testing specific hypotheses, and how an experimentalist could prioritize specific hypotheses to test.

Thank you for the suggestion. The Discussion section now contains a complete paragraph suggesting experimental follow-ups in three directions: (i) direct flux measurements; (ii) genetic perturbation studies; and (iii) ecological studies, all focusing on the most pronounced predicted effects of habitat, physiology, and taxonomy on metabolic network operation.

Minor comment

Fig. 2b: it is a bit tricky to resolve curves and colors. I understand that many curves may be overlapping, with little that can be done to improve readability. But it should be pointed out in the legend which curves are overlapping. Also, two curves have very similar green color, which may be a source of ambiguity.

We have now increased contrast (line for EC score four is in black) and added a comment on overlapping curves to the caption.

Reviewer #2 (Remarks to the Author):

The manuscript by Ramon and Stelling summarises the results of inter-species comparisons between metabolic networks based on a stoichiometric matrix-based metric (previously published by the last author). The metric assesses similarities in the sensitivities of the networks in question in response to a perturbation at shared reaction nodes. Compared to commonly used metabolism comparison metrics (e.g., Jaccard distance or EC mapping similarities), the metric used here accounts for the structural dependencies between reactions within a network. Thus, the comparison is much more meaningful and hints at structural conservation (or lack thereof) of the networks in question. While I generally find the approach very useful, there are several points where the study and the manuscript is unclear / requires improvement.

We thank you for the concise summary and for the positive comments on the metric we propose.

1. The use of ‘function’ and ‘functional’ in the manuscript is very confusing and often inaccurate. I think ‘Structural comparison’, for example, would be a much more accurate description. The function is generally, and rightfully, thought of as a phenotype or contribution to a, say, biochemical pathway. Perturbation sensitivity of a reaction does not inform much about the function of that reaction or organism per se; two networks could be very different their sensitivity profiles but could be functionally similar under a set of environments (consider two organisms with each having a different set of parallel pathways).

We appreciate the reviewer’s point and pondered this aspect of terminology quite extensively in preparing the manuscript. We came to the conclusion that ‘structural comparison’ would be a less adequate term because our approach does not compare network structures (i.e., topologies), but metabolic phenotypes (predicted flux perturbations). It does so with two important qualifications: (i) the predictions are derived from network structure alone and do not depend on assumptions on the environment (please see the response to reviewer #1, point 1 for details), and (ii) the analysis is differential (quantifying sensitivities of predicted flux distributions to perturbations, and not predicted flux distributions by themselves, which makes the metric independent of the environment).

We think that the example of two organisms with a different (in terms of pathway stoichiometries) set of parallel pathways is very illustrative for the implications of our approach. In this scenario, if the stoichiometric interactions of the two parallel pathways with the rest of the metabolic network in the two organisms were identical, sensitivity correlations for the two organisms would be the same (and one if the rest of the metabolic networks were identical) because we compute the correlations based on the common reactions (excluding the parallel pathways in this case). Hence, sensitivity correlations would reflect the functional similarity stated by the reviewer.

To improve accuracy and avoid confusion as early as possible, we now start the Introduction (1.26) as follows: “Metabolic reactions as well as entire metabolic networks establish function by yielding phenotypes in terms of metabolic flux distributions inside the cell and in the cell’s interaction with the environment. Understanding the drivers of metabolic functional diversity requires disentangling links between metabolic gene repertoires, realized metabolic phenotypes, taxonomy to represent evolutionary history, and environmental characteristics.” We hope that this makes the more specific statement (1.62) “Correspondingly, we use ‘function’ (‘functional similarity’) in the sense of (similarity of) phenotypic responses to

perturbations of the common reactions.” clearer. In addition, we checked all occurrences of ‘function’ / ‘functional’ in the manuscript for consistency with these definitions.

2. The method section on sensitivity calculation is very incomplete. While I appreciate that the method is published previously, there are several details that are important: in particular, under what media conditions were the sensitivity calculated? It would also be important to put the method in the context of other widely used GSM-based metrics such as flux variability analysis.

Thank you for this suggestion; we have now expanded the corresponding Methods subsection to provide context as follows:

“Structural sensitivity analysis¹⁵ quantifies how the perturbation of a reaction flux affects all fluxes in a GSM, assuming minimal total flux adjustment as in the minimization of metabolic adjustment (MOMA) method⁵³. Specifically, we computed absolute structural sensitivities (see **Supplementary Methods** for details). They neither require the definition of a reference flux distribution (as in MOMA⁵³), nor the definition of an environment through constraints on exchange fluxes (as in flux variability analysis, FVA⁵⁴). Hence, absolute structural sensitivities characterize network responses independent of a cell’s metabolic state or environment.”

In addition, we have stated early on in the main text that our sensitivity calculations do not assume any specific media / environment; please see the responses to the other reviewers’ corresponding comments for details.

3. Figure 2. (Note that there is no panel d as the caption mentions.) The panel (b) seems like a circular argumentation, EC mappings are used in GSM reconstruction, so it is expected that sensitivity similarities would reflect gene similarities.

Thank you for spotting the wrong reference to panel d – we have corrected this.

Regarding enzyme similarities, we see that our previous statements on Fig. 2b lacked sufficient detail. Indeed, it is expected that enzymes with very similar EC classification would have very similar metabolic function (up to catalyzing identical reactions, which is exploited in GSM construction) when seen in isolation. However, when accounting for the propagation of perturbations through different (yeast and human) networks via sensitivity similarities our data shows that (i) reactions with identical EC numbers do not necessarily have a large sensitivity correlation, reflecting that they operate in different network contexts, and (ii) EC numbers with at least one different number do not show any similarity in sensitivities, indicating an enzyme’s position in the network rather than the coarse chemical reaction class it catalyzes relates to its effect on network operation.

The revised text summarizing these arguments (l.100) now reads, “As expected, enzyme pairs with the highest EC number similarities showed higher sensitivity correlations than more unrelated pairs (Fig. 2b). However, enzymes with identical EC numbers do not necessarily have high sensitivity correlations, reflecting their different network contexts. This context-dependence dominates over coarse classification of catalyzed chemical reactions because even a single difference in EC number abolishes correlations.”

4. The overall message/conclusion is unclear: are the authors proposing a new metric for pathway/subsystem identification? The conclusion that the framework can help in joint

interpretation of traits, evolution and environment is quite vague, especially with a fuzzy definition of trait (similar discussion as above for function).

Regarding terminology, we previously used ‘metabolic phenotype’ and ‘metabolic trait’ synonymously. We have now replaced ‘trait’ by ‘phenotype’ in all but one case where ‘trait’ is the term used in the cited reference.

In general, we have substantially expanded and revised the Conclusion (now Discussion) section to make the overall message clearer. For example, we now state on 1.336 that “[w]e consider the concept of sensitivity correlations as the main contribution of this work, and the range of applications presented as proofs-of-principle.”

Reviewer #3 (Remarks to the Author):

Ramon and Stelling introduce the application of structural sensitivity analysis as a way to compare the function of metabolic networks across organisms. Instead of comparing the composition of metabolic networks across species or values of specific traits inferred from corresponding models, the authors focus on the similarity of responses to perturbations of individual reactions between organisms. They use this framework to, among other proof-of-principle applications, assess the contribution of taxonomy and the environment to the functional differences inferred by their method. I believe the approach presented is novel and potentially useful. The methods used are mostly appropriate and support the main claims of the manuscript. The main points to revise relate to the clarity in the interpretation of the functional conservation/variability examples, and the robustness of the metric to metabolic model completeness.

Importantly, the comparison of metabolic phenotypes inferred from metabolic reconstructions with regards to taxonomic or environmental classifications is not a novel concept. For example, Bernstein et al (<https://elifesciences.org/articles/39733>) predicted the biosynthetic capabilities of 456 oral microbes and, similar to the present study, defined a functional distance between microorganisms that was associated with taxonomy and co-occurrence data. Zarecki et al. (<https://dx.plos.org/10.1371/journal.pcbi.1003726>) used the metabolic reconstructions of 2,529 microorganisms to predict minimum growth requirements for growth and compared those predictions with corresponding ecological and genomic traits. Therefore, the novelty of the current study is mainly about 1) the nature of the measure of functional similarity proposed and 2) the focus on the relative contribution of evolutionary history versus environment to the variability of said measure.

Thank you very much for your detailed and highly constructive comments.

Specific comments

1. Line 13. In the abstract the sentence "model-predicted traits were only employed at the species level" should be revised in light of the above references. Similarly, for Line 38, although reference 6 did not make links to genotype or environment, the above papers that did should be cited.

We apologize for the inaccuracies and thank you for pointing out the two references we missed. The sentence in the abstract now reads "... and rarely were model-predicted phenotypes employed beyond the species level⁶⁻⁸" including the above references. Correspondingly, we have re-formulated the Introduction to "A recent, comparative GSM-based study of bacterial phenotype evolution⁶ did not make links to genotype or environment, while other comparative studies linked specific (minimal)⁸ or generic (sampled)⁷ environments to species and ecological relations, but not to detailed metabolic functions."

2. Line 58. For the comparison between sensitivities and Jaccard similarities, the size of neighborhoods used to obtain the results in Fig 1b is not specified. This threshold is only defined as delta in the supplementary methods. Thus, it is not clear whether the size of these neighborhoods affects the correlation between the two measures.

We have added the size of the neighborhood (1-neighborhood) to the caption of Fig. 1b and in the main text. To assess the effect of the size of the neighborhood, we computed Jaccard indices for 2-neighborhoods (note that the maximum graph distance in the two networks is 5); the conclusion on the lack of correlation does not change (new Fig. S1b and main text).

3. Fig 1d. I believe the description of the solid and dashed red lines in the figure legend are inverted. Please check.

Thank you for spotting this mistake – we corrected it.

4. Line 85. The analysis shows that reactions associated with orthologous genes or reactions with the same EC number have higher sensitivity correlations than non-orthologous reactions and reactions with different EC numbers. This suggests that identical reactions are not always functionally equivalent across species, but does not necessarily validate reaction-level predictions, as the authors claim at the start of the paragraph. This is because reactions with EC scores 0-3 as well as non-orthologs all have the same low sensitivity correlation. Instead, the authors should look at the sensitivity correlations of reactions within the same pathway, or reactions at increasing distances in the network graph. As the main point would be that functionally related reactions have higher sensitivity correlations.

Regarding the start of the paragraph, we have revised the text to avoid a potentially misleading statement on validation (please see also response to reviewer #1, point 3).

Thank you for the detailed proposal for alternatives. Our analysis already addresses these aspects: first we show that sensitivities do not correlate in general with reaction distance with a few examples (Fig. S1a), highlighting that reaction distance is not a good predictor of the magnitude of the effect of responding to a perturbation (a reaction can be far away from the perturbation in terms of distance but respond strongly to a perturbation because it is coupled through a linear pathway, for example). More fundamentally, we agree with the expectation that ‘functionally related reactions have higher sensitivity correlations. This is why we examined the sensitivity correlations of reactions within the same pathway in Fig. 2a for *E.coli* and *B. subtilis*. We then expanded this analysis in Fig. 4: sensitivity correlations show large within-pathway variations that are pathway-specific, as shown in Fig. 4a,b.

Please note that reaction classifications for pathways may become very ambiguous, considering the limited overlap of pathway classifications in, e.g., SEED and KEGG databases. Pathway analysis is further complicated by ‘traditional’ metabolic pathways not capturing all reactions belonging to steady-state flux distributions – from formal pathway analysis (e.g., via elementary flux modes), we know that functional pathways are usually substantially larger than ‘traditional’ pathways.

We hope that (i) reaction-level validation of predictions is now more clearly covered by the revised section on functional alignment (please see our response to reviewer #1, point 4 for details), and (ii) the additional explanations in response to your comments below clarify the more general relations of pathways and functions.

5. Line 101. For clarity, it should be specified that the number of common reactions refers to the reactions used to calculate the sensitivity correlations as opposed to the number of common reactions in the models used to calculate the sensitivities.

Thank you for this suggestion. We have re-phrased the sentence to: “Also, reducing the number of reactions to calculate sensitivity correlations had only a small effect ...”

6. Line 112. Same as above, it should be made clear that common reactions refers to the number of reactions used to calculate the correlations. Not common reactions in the models.

Similar to the previous point, we have modified the statement to: “In contrast, more than 92% of the metabolic reactions were correctly aligned even when using only 1% of the reactions to compute sensitivity correlations (Fig. 3a).”

7. Line 148. Why are normalized biases calculated relative to the regression line between reaction usage and sensitivity correlations (as described in the supplementary methods) if there is no correlation between those two variables according to Figure S4c? Would the same results be obtained if a single mean and standard deviation across all reactions was used for normalization?

Indeed, with the lack of correlation (which we think is important to consider and show), normalized biases amount to z-scores. We have now clarified this in the main text (l.171, “We quantify functional similarity across multiple bacteria using normalized biases (z-scores, that is, standard deviations from the mean) ...” and added an explanation to the SI “Without significant correlations between reaction usages and sensitivity correlations in our data (Fig. S4c), these definitions amount to computing z-scores, as stated in the main text.” We kept the mathematical formulation to retain generality and consistency of notation.

8. Lines 165-176. Though these are interesting observations, it is unclear how the authors are interpreting processes classified as conserved or variable. For instance, the fact that there's only one universal histidine biosynthesis pathway does not necessarily imply that such pathway is functionally conserved based on sensitivity correlations. Specifically, the result also indicates that the reactions most affected by fluxes through the histidine pathway are similar across species. Perhaps the authors can highlight with specific examples that the sensitivity results are not always explained by the universality of the pathways but by the processes they influence across the whole network.

Thank you very much for this excellent suggestion – we have re-written and expanded this and the preceding paragraph to provide clearer interpretations and examples.

9. Line 202. Figure 4b. The normalized biases calculated for the nodes of the environment trees are very small (close to 0), so the color scheme does not easily allow seeing the patterns described in the text. I suggest adding a different representation, or supplementary figure with the mean biases for each of the environmental categories.

We augmented Figure 4b by including the numerical values of (significant) regression coefficients for normalized biases, and modified the caption accordingly.

10. Line 202. In general, the results in this paragraph are hard to understand. The authors start by saying that host environments associate with low functional variability compared to water and soil, but then say the opposite result is true after controlling for other factors? which factors? where are these results shown in the figures? In the same paragraph, the authors say that bacteria in low-temperature habitats are functionally conserved compared to thermophiles and that this is consistent with reference 41 that found the metabolic networks of bacteria living at high temperatures to be less modular and less flexible. How is this lack of flexibility consistent with more functional variability? Shouldn't the opposite be true?

First, we realized that the caption for Figure 4b was incomplete, and hope that the revision clarifies what data is shown.

For environmental influences, we revised the text to separate the two results / arguments of (i) similar environments leading to similar functional distances, and (ii) environmental variability

alone not dictating functional variability (e.g., because of resource constraints that can enforce functional conservation). Regarding the influence of temperature, we corrected the misleading ‘flexible’ terminology and now explain the rationale to link modularity and metabolic variability.

11. Line 240. The authors say that conserved siderophore metabolism could enable cooperation, but then hypothesize that functional variability facilitates the evolution of cooperation. For this and the point above, the authors should devote some space to explain how they are interpreting their results for the examples provided and not just point to possibly relevant publications, as these connections are not obvious.

Thank you for the suggestions to explain the examples in more detail; we modified the text accordingly. Regarding the evolution of cooperation, we now make more detailed arguments – especially regarding cooperation via siderophores, which can be considered an exception because of the specific physico-chemical constraints of the marine environment. Not considering such constraints in our analysis approach is a limitation that is now stated.

12. Line 254. It is important to provide an estimate of the impact of model errors on the predictions by the proposed method. Would the obtained results change significantly for different metabolic models for the same species (e. g. generated via different methods)? Based on the results in figure S3b, would a similar strategy: deleting different numbers of random reactions from selected models, indicate the robustness of the sensitivity correlation metric towards model ambiguities or incompleteness? I think the authors can easily comment and quantify this by applying that strategy to other models.

For a discussion of effects of model errors (or different coverage or accuracy for GSMs of the same species), please see our response to reviewer #1, point 7.

We appreciate the suggestion of an extended robustness analysis, which we have implemented for the analysis of the microbial GSMs (results in Fig. 4) as follows: To approximate model inaccuracies and varying coverage (e.g. of common reactions due to annotations), we subsampled sensitivity correlations and assessed the effect on results presented in Fig. 4b, namely (i) clustering relations between taxonomy, habitat and physiology (THP) variables (new Fig. S11) and (ii) hypotheses on significant subsystem variability or conservation (new Fig. S12). Similar to the data in Fig. S3b, these results demonstrate high robustness of the predictions discussed in the main text to model variation, and we added corresponding statements to the subsections ‘Drivers of metabolic diversity’ (l.238) and ‘Diversity of metabolic subsystems’ (l.280).

13. Line 282. It would help for the authors to specify that the sensitivities calculated are absolute sensitivities, and not relative sensitivities, such that the analysis does not require that reference fluxes or simulation conditions are specified. A sentence early in the manuscript clarifying this would facilitate understanding of the methods and interpretation of the results.

Thank you for this excellent suggestion, also given the other reviewers’ comments. We have included the following statement at the start of the Results section (l. 54): ‘Here, we computed absolute sensitivities, which do not assume a specific operating state of the network or a specific environment¹⁷ These sensitivities are based on infinitesimal perturbations, which are too small to be affected by specific constraints such as the availability of nutrients in a specific environment.’ Please see also the responses to the corresponding comments by the other reviewers.

Hoping that this helps more clearly communicate this research,
Germán Plata

Reviewer #1 (Remarks to the Author):

The author did overall a good job at addressing my concerns. Some final recommendations (mostly related to my previous first comment):

The revision of the first part of the introduction helps clarify to some extent the approach used. However, the key sentence "Specifically, sensitivities measure the effect of perturbing one reaction by predicting adjustments to all fluxes to return to steady-state in the perturbed network" can still be improved, both in terms of syntax and clarity.

Also, I now realized that I had missed some aspects of the concept of "absolute sensitivities", and think that the notion will still be a bit obscure to most readers. I think it would be very beneficial if the authors could add a couple of sentences at the beginning of the results to provide a little more accurate detail and simple intuition. For example, in the introduction they mention that they perturb the enzyme. Later in the results they mention that they perturb the reaction. Neither is accurate, as they truly perturb the flux.

I would also expect that readers may still be confused (as were at least two reviewers) on how one can talk about predictions and perturbations of specific fluxes without environmental constraints (now standard in many FBA-related calculations). Again, I would encourage the authors to spend a few lines really explaining as clearly as possible the intuition of this concept. The new sentence "These sensitivities are based on infinitesimal perturbations, which are too small to be affected by specific constraints such as the availability of nutrients in a specific environment" is actually quite confusing to me. Environmental constraints would determine the possible value of a given flux, which I could then perturb by a small amount. I would still have to perturb the flux relative to that amount, so the fact that the perturbation is small does not automatically make the flux value irrelevant.

I would also personally prefer to see the description of this crucial concept in the main paper, rather than in the supplementary material, if space constraints allow it.

Reviewer #2 (Remarks to the Author):

Overall, the authors have addressed the previous comments. I am, however, not convinced by the usage of "function" and trait/phenotype as they are too broad and to some extent misleading in this context. Consider, for example, "flux sensitivity" and "flux phenotype".

Reviewer #3 (Remarks to the Author):

After reading the revised manuscript and authors' responses, I consider the points I raised to be sufficiently addressed.

RESPONSE TO REVIEWERS' COMMENTS

Reviewer #1 (Remarks to the Author):

The author did overall a good job at addressing my concerns. Some final recommendations (mostly related to my previous first comment):

The revision of the first part of the introduction helps clarify to some extent the approach used. However, the key sentence “Specifically, sensitivities measure the effect of perturbing one reaction by predicting adjustments to all fluxes to return to steady-state in the perturbed network” can still be improved, both in terms of syntax and clarity.

We have revised this key sentence to improve syntax and clarity.

Also, I now realized that I had missed some aspects of the concept of “absolute sensitivities”, and think that the notion will still be a bit obscure to most readers. I think it would be very beneficial if the authors could add a couple of sentences at the beginning of the results to provide a little more accurate detail and simple intuition. For example, in the introduction they mention that they perturb the enzyme. Later in the results they mention that they perturb the reaction. Neither is accurate, as they truly perturb the flux.

We have added more explanations to the final paragraph of the Introduction and the first paragraph of Results. For example, we added an explanatory sentence for structural sensitivity analysis, we clarified the relations between enzymes and fluxes, which are related by gene-protein-reaction mappings, and we elaborated on absolute sensitivities / environmental constraints (see next point).

I would also expect that readers may still be confused (as were at least two reviewers) on how one can talk about predictions and perturbations of specific fluxes without environmental constraints (now standard in many FBA-related calculations). Again, I would encourage the authors to spend a few lines really explaining as clearly as possible the intuition of this concept. The new sentence “These sensitivities are based on infinitesimal perturbations, which are too small to be affected by specific constraints such as the availability of nutrients in a specific environment” is actually quite confusing to me. Environmental constraints would determine the possible value of a given flux, which I could then perturb by a small amount. I would still have to perturb the flux relative to that amount, so the fact that the perturbation is small does not automatically make the flux value irrelevant.

We have expanded / modified this section to clarify that (i) the sensitivities are computed analytically based solely on the network structure (i.e., stoichiometric matrix), without involving numerical optimizations as in most FBA-related calculations, and therefore do not require / account for a reference flux distribution; and (ii) because they do not incorporate constraints such as environment (or reaction reversibility), the sensitivities do not depend on them; sensitivities are nonetheless valid unless the unknown flux distribution for the unperturbed network lies exactly on one of the constraints, which is very unlikely in the high-dimensional flux spaces considered in genome-scale metabolic models. We hope that the abbreviated version of this rationale in the revised first paragraph of Results is clear now.

I would also personally prefer to see the description of this crucial concept in the main paper, rather than in the supplementary material, if space constraints allow it.

With the integration of Supplementary Methods into the Methods section, the detailed description is now in the main paper.

Reviewer #2 (Remarks to the Author):

Overall, the authors have addressed the previous comments. I am, however, not convinced by the usage of "function" and trait/phenotype as they are too broad and to some extent misleading in this context. Consider, for example, "flux sensitivity" and "flux phenotype".

We checked every instance of 'function' / 'functional' and considered replacements as suggested. However, either the use of these terms is required in the broad sense (e.g., in Abstract and Introduction to discuss prior work, open problems), or in the narrow sense of 'function characterized by flux sensitivity' to contrast with 'structure' or 'metabolic repertoire'. For the latter cases, we feel that with the defined terminology in the context of the Results section, direct replacements would make the text more incomprehensible while not being necessary. We therefore left this terminology unaltered.

Similarly, for 'phenotype', we considered where this general term is appropriate, for example, in referencing or otherwise relating to prior work, and where it is specific to our framework. With this rationale, we kept 'phenotype' in Introduction and Discussion as well as one instance of Results (where the text explains an experimental study) and replaced it by 'flux phenotype' elsewhere (affecting only the Methods section because we already made corresponding modifications in the first revision).

Reviewer #3 (Remarks to the Author):

After reading the revised manuscript and authors' responses, I consider the points I raised to be sufficiently addressed.

We thank the reviewer for the positive evaluation.